# Deletion of Mediator 1 suppresses TGFβ signaling leading to changes in epidermal lineages and regeneration

Yuko Oda[1]*, Thai Nguyen[1], Akiko Hata[2,3], Mark B. Meyer[4], J. Wesley Pike[4], Daniel D. Bikle[1]

**1** Departments of Medicine and Endocrinology, University of California San Francisco and Veterans Affairs Medical Center San Francisco, San Francisco, CA, United States of America, **2** Cardiovascular Research Institute, University of California, San Francisco, CA, United States of America, **3** Department of Biochemistry and Biophysics, University of California, San Francisco, CA, United States of America, **4** Department of Biochemistry, University of Wisconsin-Madison, Madison, Wisconsin, United States of America

\* yuko.oda@ucsf.edu

**Data Availability Statement:** The microarray and Chip-seq data were submitted to a public database (GEO/NCBI/NIH http://www.ncbi.nlm.nih.gov/geo). Data for Med1 KO epidermal cells (10 wk), KO

## Abstract

Epidermal lineages and injury induced regeneration are controlled by transcriptional programs coordinating cellular signaling and epigenetic regulators, but the mechanism remains unclear. Previous studies showed that conditional deletion of the transcriptional coactivator *Mediator 1 (Med1)* changes epidermal lineages and accelerates wound re-epithelialization. Here, we studied a molecular mechanism by which Med1 facilitates these processes, in particular, by focusing on TGFβ signaling through genome wide transcriptome analysis. The expression of the TGF ligands (Tgfβ1/β2) and their downstream target genes is decreased in both normal and wounded Med1 null skin. *Med1* silencing in cultured keratinocytes likewise reduces the expression of the ligands (TGFβ1/β2) and diminishes activity of TGFβ signaling as shown by decreased p-Smad2/3. Silencing *Med1* increases keratinocyte proliferation and migration *in vitro*. Epigenetic studies using chromatin immuno-precipitation and next generation DNA sequencing reveals that Med1 regulates transcription of TGFβ components by forming large clusters of enhancers called super-enhancers at the regulatory regions of the *TGFβ* ligand and *SMAD3* genes. These results demonstrate *that Med1* is required for the maintenance of the TGFβ signaling pathway. Finally, we show that pharmacological inhibition of TGFβ signaling enhances epidermal lineages and accelerates wound re-epithelialization in skin similar to that seen in the *Med1* null mice, providing new insights into epidermal regeneration.

## Introduction

Skin epithelia are highly regenerative tissues developmentally derived from the ectoderm. They are differentiated into three appendages, interfollicular epidermis (IFE), sebaceous glands (SG) and hair follicles (HF). They drive lineage specific transcriptional programs potentially by coordinating cell signals and epigenetic regulators. These lineages are supported by

whole skin (10 wk), and KO skin wounds (1d after wounding) are available with accession numbers GSE50672, GSE50671, GSE50670, respectively under the super-series GSE50673. The ChIP-seq data for keratinocytes immunoprecipitated with H3K27ac, CTCF and Med1 antibodies are available with accession number GSE154221.

**Funding:** This work was supported by the NIH grant R21 DE025357 (YO), R01 AR050023 (DDB), DOD grant CA110338 (DDB), VA Merit I01 BX003814-01 (DDB).

**Competing interests:** The authors have declared that no competing interests exist.

**Abbreviations:** Med1, Mediator 1; Krt, keratin; Flg, fillagrin; Lor, loricrin; TGFβ, transforming growth factor beta; HF, hair follicle; IFE, interfollicular epidermis; SG, sebaceous gland; Isthmus/JZ/I, the isthmus and junction zone and infundibulum; PPARγ, peroxisome proliferator activated receptor γ; ChIP, chromatin immunoprecipitation and DNA sequencing; SE, super-enhancers; TF, transcription factor; ESC, embryonic stem cells; KO, knockout; CON, control; IPA, Ingenuity pathway analysis; ECM, extracellular matrix; d, day; wk, week; RS, RepSox; α-SMA, α-smooth muscle actin.

adult stem cells and their progeny residing in distinct niches. The epidermis is regenerated by putative epidermal stem cells in the basal layer [1] which with differentiation produce epidermal markers such as Keratin 1 (KRT1), Involucrin (IVL), Loricrin (LOR), and Filaggrin (FLG) in part utilizing AP-1 factors of the Fos/Jun families. Sebocytes are derived from isthmus/junctional zone/infundibulum (isthmus/JZ/I) in upper hair follicle (HF) [2]. *Prdm1/Blimp1* is key regulator within sebocytes in their production of lipids driven by PPARγ [3]. Hair is regenerated from the lower bulge region of HFs [1], the regulation of which is under the control of a number of transcription factors including those of the hedgehog and wnt/β -catenin pathways. Under normal circumstances, these lineages are controlled by distinct transcriptional programs, but upon injury, the stem cells and progeny near the wound site alter their fate to contribute to the re-epithelialization of the wounds. However, the master regulator governing these lineages and enabling their response to injury is not known, limiting the development of a strategy to improve epidermal regeneration during cutaneous wound healing.

Mediator is a transcriptional coactivator recently implicated in cell specific transcription. Mediator was originally thought to facilitate nuclear receptor or transcription factor driven transcription as part of the RNA polymerase II complex [4–6]. For example, we previously identified the Mediator complex originally known as VDR interacting proteins (DRIP) as critical for the actions of the vitamin D receptor (VDR) in epidermal keratinocytes [7]. However, epigenetic studies *(in vitro)* have expanded this concept by showing that Mediator forms large clusters of enhancers called super-enhancers (SEs) [8, 9]. SEs are implicated in cell fate determination and control lineage plasticity [10] by facilitating cell specific transcription by linking them to the initiation complex at these specific promoters [11]. For example, Mediator facilitates the transcription of four key transcription factors (TFs) that reprogram somatic cells to embryonic stem cells (ESC) through SEs, whereas the reduced levels of Mediator induce ESC differentiation by blunting the activity of these TFs [8, 12].

Previously, we evaluated the function of one of the subunits of the Mediator complex, *Mediator 1 (Med1)* (also called DRIP205 or TRAP220) [4–6]) in somatic cell fates. Conditional deletion of *Med1* from Krt14 expressing epithelia postnatally switches the cell fate of dental epithelia to that generating hair [13–15]. In contrast, the same *Med1* KO mice show hair cycle defects resulting in hair loss (alopecia), but with an increase in IFE markers [13]. Moreover, Krt5 driven *Med1* KO mice were noted to have accelerated wound re-epithelialization at 8 wk [16]. These results suggest that *Med1* governs epidermal lineages in skin and teeth.

In this study, we explored the molecular mechanisms by which *Med1* facilitates these processes through genome wide transcriptomic and epigenetic analyses followed by *in vivo* functional studies. We focused on the transforming growth factor β (TGFβ) signaling because our current genome wide transcriptomic studies presented in this report indicate that TGFβ is a top candidate for mediating *Med1* action. In addition, TGFβ signaling is known to participate in the functions of various stem cells [17].

TGFβ ligands (TGFβ 1, 2) bind to the type I receptor (TGFbR1 also known as Alk5) and type II receptor (TGFbR2). Upon ligand binding to the type I receptor, transcription factors of SMAD2 and SMAD3 are phosphorylated and induce transcription of target genes [18, 19]. TGFβ signaling regulates production of extracellular matrix (ECM) proteins such as collagens and connective tissue growth factors such as *CTGF*/CCN2. TGFβ maintains quiescence of epithelial stem cells and supports cell lineages by functioning as a specific master regulator but in a context dependent manner [17]. The activity of TGFβ signaling can be targeted by drugs to treat various diseases [20]. One of them is RepSox, that blocks TGFβR1/TGFβR2 phosphorylation. We used this drug as a pharmacologic test of the role of TGFβ signaling in the maintenance of epidermal lineages and regeneration during cutaneous wounding healing.

In the current study, we show that *Med1* regulates TGFβ signaling through epigenetic regulation of TGFβ ligands and its transducer transcription factor (TF) (SMAD3). Our results suggest a means of enhancing injury induced epidermal regeneration by manipulating TGFβ signaling.

## Materials and methods

### Krt14-driven *Med1* KO mice

Conditional *Med1* KO mice were generated as described [13]. Briefly, floxed (exon 8–10) *Med1* mice [21] (C57/BL6 background) were mated with transgenic mice expressing *Cre* recombinase under the control of the keratin 14 (Krt14) promoter (The Jackson Laboratory, C57/BL6 background). Genotyping was performed by PCR as previously described [13]. The Cre negative littermate mice serve as controls (CON). All of the experiments were approved by the Institutional Animal Care and Ethics Committee at the San Francisco Department of Veterans Affairs Medical Center.

### Microarray analysis

Gene expression profiles were analyzed using a lllumina beads chip based gene array (Mouse Ref-8 v2.0 Ambion) including 25,600 annotated transcripts and 19,100 genes. Sample preparation, labeling and array hybridizations were conducted according to standard protocols by the UCLA Neuroscience Genomics Core Facilities.

The data were normalized in the Genome Studio (Illumina). The fold changes (log) in gene expression of *Med1* KO over control were calculated (average intensity, *p*-values, and standard deviation). The data were then analyzed by using the Ingenuity IPA software (Ingenuity) via data transformation by setting the score and the *p*-value less than 0.005. Several analyses performed to predict upstream regulators and affected pathways were conducted using IPA software. The upstream molecules were listed, which are potentially responsible for the observed changes in gene expression. The statistical significance of potential upstream regulators was evaluated by the z-score and the *p*-values calculated through the algorithm installed in IPA software. Heat maps were prepared by using MeV software using fold changes (KO/CON).

### Chromatin immunoprecipitation and DNA sequencing

Chromatin IP was performed by using the LowCell# ChIP kit (Diagenode) for H3K27ac, and by using the iDeal ChIP-kit for transcription factors (Diagenode) for MED1 and CTCF according to the manufacturer's protocol with some modifications as described.

Pre-confluent keratinocytes were cross-linked by 1% paraformaldehyde for 8 min for H3K27ac and for 15 min for MED1/CTCF then quenched with 0.125M glycine. Whole cell lysates (for H3K27ac) or purified chromatin (MED1/CTCF) were sonicated by Covaris S2200 ultrasonicator (Covaris, Inc.). The shearing conditions were optimized to obtain the DNA fragments with to an average length of 300±50 bp, and the size of which was verified by an Agilent 2100 Bioanalyzer (Agilent Technologies). Sheared chromatin was immunoprecipitated with Protein A-coated magnetic beads (Diagenode) preincubated with antibodies: 3 μg of antibody against H3K27ac (ab4729, Abcam), 1 μg CTCF antibody (Diagenode) and 4.5 μg MED1 antibody (Bethyl). Experiments were carried out in duplicates (2) for MED1, in triplicate for CTCF (3), and once for H3K27ac (1), and a chromatin input sample was used as a reference to subtract background for peak calling. Complexes were washed, eluted from the beads with washing buffer and crosslinks were reversed by incubation at 65C for 4 hr.

Immunoprecipitated DNA along with genomic DNA (Input) were purified using IPure v2 kit (Diagenode). IP efficiency was confirmed by qPCR using primers provided in the kits.

ChIP-seq library preparation: DNA Sequencing libraries were generated using the Accel-NGS 2S Plus DNA library kits (Swift Sciences) and amplified by PCR for 11 cycles. To remove high MW smear in the library, right side size selection was conducted by using SPRI beads (Beckman Coulter). The library was quantified by Agilent 2100 Bioanalyzer with the High sensitivity DNA assay. They were sequenced on an Illumina HiSeq 4000 (UCSF Center of advanced technology) in a single strand 50bp run (combining 8 libraries per lane).

Sequencing reads were mapped to the human genome hg19 using Bowtie2 with standard parameters only allowing uniquely aligned reads as previously described [22–26]. Peaks were called against input using MACS2 for H3K27ac, Med1, and CTCF using both broadPeak and gapped Peak output [27]. Motif analysis (de novo and known) was performed using HOMER software [28]. All data were displayed using the UCSC genome browser. Enhancers were stitched and super-enhancers were defined using ROSE code as described [8]. ROSE was run with stitching distance of 12.5kb. SE were then assigned to the RefSeq genes whose TSS was the nearest to the center of the identified SE deposited in the Gene Expression Omnibus.

### Reproducibility and statistical analysis

All of the experiments using *Med1* KO mice were repeated with at least two litters, and reproducibility was confirmed. The experiments using cultured keratinocytes were conducted in triplicates (3 wells) for statistical analyses, and reproducibility was confirmed by using independent batches of cells. Histologic images were quantitated by Bioquant software using 2 sections per mouse with 3 mice per each group of KO and CON. Statistical significance was calculated using the Prism software 6 (GraphPad Software, Inc.) or in two-tailed unpaired Student's t-test available in Excel. Differences with $p$-values of less than 0.05 were considered statistically significant and indicated by asterisks.

Other experimental procedures are described in Supporting information.

## Results and discussion

### *Med1* ablation reduces expression of TGFβ ligands, Smad3 transducer and TGFβ downstream target genes

To elucidate a molecular mechanism of *Med1* action, we determined genome wide transcriptional profiles through microarray analyses. *Med1* conditional KO mice, in which the *Med1* gene was excised by Cre recombinase driven by the Keratin (Krt) 14 promoter, and littermate Cre negative control (CON) mice [13, 15, 21] were used. Epidermis was separated from dermis in *Med1* KO and CON mice at 10 wk of age when abnormal anagen shift was obvious in Med1 KO skin compared to CON skin (telogen). Epidermal keratinocytes were isolated from epidermis with mild trypsin treatment to exclude differentiated cells. Total RNAs were purified and applied to a bead-based gene array as previously described [14]. Expression profiles of KO and CON keratinocytes were compared (GSE50672 n = 4 under super series GSE50673). Upstream analyses using IPA software listed potential regulators through which Med1 functions. Both the protein and the mRNA for TGFβ 1 came to the top of the list with greatest statistical significance (activation z-scores with p-values) (S1 Table red letters) among other candidates (S1 Table). In addition, they were predicted as inhibitory regulators (S1 Table fourth column). The TGFβ target genes were also reduced in KO cells (Fig 1A heat map, 1B pathway map). These are mainly genes for ECM proteins such as *Col6a1*, *Col3a1*, *Col17a1*, and ECM related genes such as *Ctgf*, *Cyr61*, *Cd34*, *Igfbp3*, in which IPA annotation suggested their role in cell

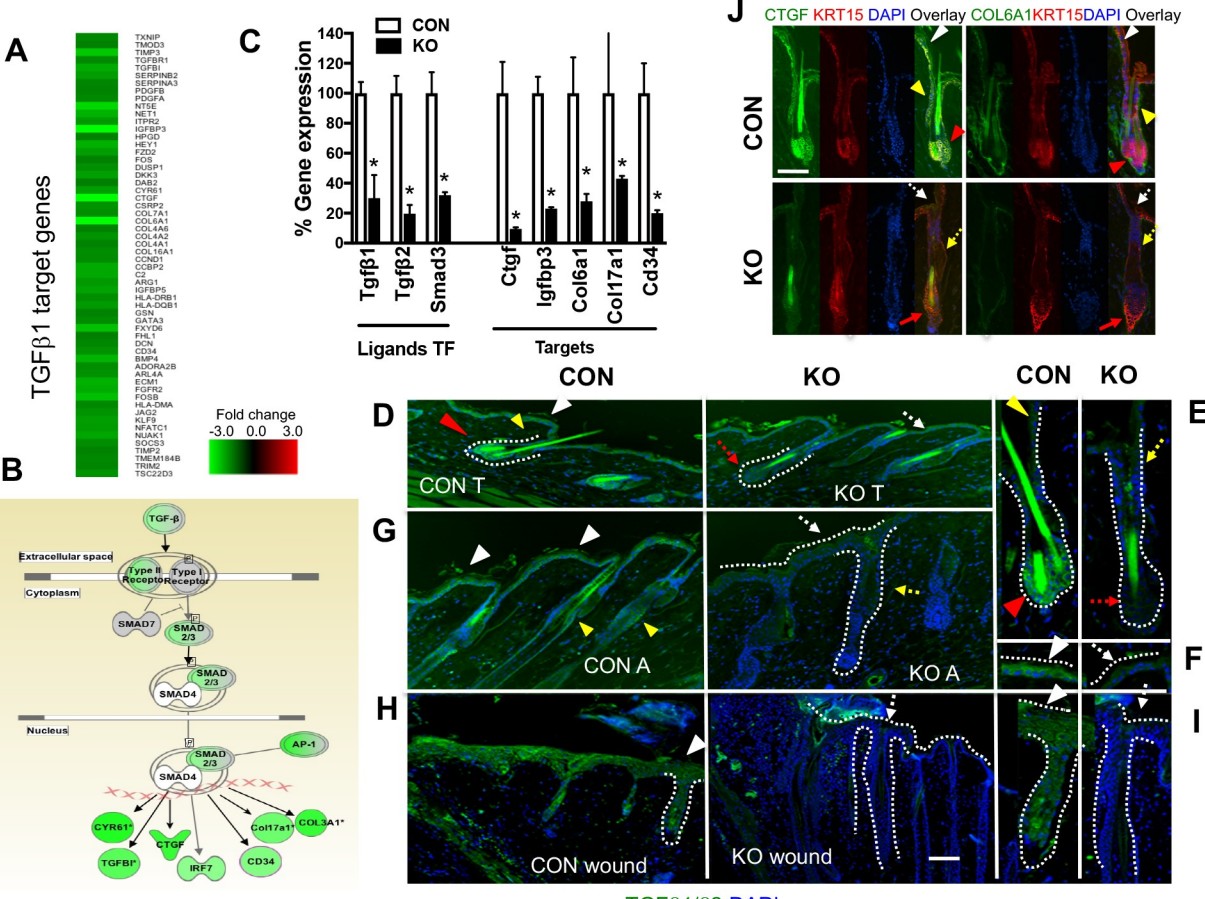

**Fig 1.** *Med1* **ablation reduces TGFβ signaling in the skin.** (A) Expression profiles of keratinocytes isolated from Med1 KO and CON mice (10wk) were compared by using microarray. The heat map (fold changes) shows that a number of TGFβ1 target genes are down-regulated in KO keratinocytes (green). (B) Pathway analysis (IPA) shows down-regulation of TGFβ pathway in KO keratinocytes. Down-regulated genes are shown by green color. (C) The mRNA levels of TGFβ ligands, signal transducer (SMAD3), and a series of TGFβ target genes in CON (open bars) and KO keratinocytes (closed bars). Percentage expression of KO compared to CON was calculated by qPCR measurements (mean +/- SD, n = 3, t-test * p<0.05). (D-I) Immuno-staining of TGFβ1/β2 in telogen (D), anagen skin (G), and wounded skin (1d) (H) of CON and KO skin. Green signals (TGFβ1/β2) overlaid with blue DAPI counterstain. Bars = 50 μm. The pictures were taken by the same exposure between CON and KO to visualize green signals in CON, that masks blue counterstain by overlay. However, the lack of green signal shows DAPI blue background in KO. The shape of HFs and edges of epidermis are shown by dotted lines. (E) Enlarged images of TGFβ1/β2 staining focused on HF in Fig D, in which HF is shown by dotted line. The locations of TGFβ1/β2 (green) are demonstrated by colored triangles; HF (red), isthmus (yellow) and IFE (white) in CON skin. Lack of the ligands in KO at equivalent locations is shown by dotted arrows by the same colors. (F) Enlarged images of TGF ligands in Fig D, focused on IEF (white arrows). (G) Immuno-staining of TGFβ1/β2 at anagen skin characterized by long HF. The location of the ligands is shown by colored triangles similar to telogen skin (D). (H) Immuno-staining of TGFβ1/β2 at wound edges of 1 d skin wounds. (I) Enlarged images of TGF ligands in Fig H, focused on HF/IFE region (shown by dotted lines) of wounded skin. (J) Immuno-staining of CTGF and COL6A1 (green, left panels), KRT15 (red, middle) and DAPI (blue, second from right), with merged images (right panels) of epidermis and HFs at 10 wk. Bars = 50 μm.

quiescence and migration (Fig 1B). The mRNA expression of *Tgfβ* ligands (*Tgfβ1/β2*) and its signal transducer Smad3 decreased in KO cells (Fig 1C qPCR, B pathway). The expression of TGFβ1/β2 proteins was substantially diminished in KO skin compared to CON (Fig 1D–1G). The location of the ligands was demonstrated by colored triangles; IFE (white), isthmus/JZ/I (yellow), and HF bulge (red) in CON skin (Fig 1D left CON). However, the levels of the ligands decreased in KO skin when CON is in telogen (T) (Fig 1D KO). Enlarged images of HF in Fig D are shown in Fig 1E, in which shape of the HF is shown by dotted lines. Enlarged IFE in Fig D, on which the area with a white triangle is focused, is shown in Fig 1F. Lack of the TGF

ligands (green) in KO skin is indicated by dotted colored arrows in IFE (white), isthmus/JZ/I (yellow), and HF bulge (red). The TGF ligands were also diminished in KO compared to CON skin at anagen as shown by long HFs (A) (Fig 1G CON KO). The presence and absence of the ligands are shown by the same colored triangles and dotted arrows, respectively. These results demonstrate that these changes are independent of hair cycling. The anti-TGFβ antibody we used recognizes both TGFβ1 and TGFβ2.

The TGFβ target proteins (green), CTGF and COL6A1, were present as shown by colored triangles in isthmus/JZ/I (yellow) and bulge (red) and IFE (white) in CON (Fig 1J upper panels), but they were diminished in KO. Lack of green signals in equivalent isthmus, bulge and IFE is shown by dotted arrows (Fig 1J KO overlay images). The KRT15 antibody (red) revealed the location of HF stem cells but also stained other keratinocytes in IFE and isthmus/JZ/I. These results demonstrate that *Med1* inhibits the expression of TGFβ signaling genes from epidermal keratinocytes. Immuno-staining of skin sections by SMAD3 and p-SMAD2/3 antibodies was inconclusive.

### *Med1* ablation reduces TGFβ ligands during injury induced epidermal regeneration

Next, we explored *Med1* regulation of TGFβ signaling during injury induced regeneration of the epidermis. We induced the wound response in skin by taking a full thickness biopsy (3 mm) from the back skin of KO and CON mice at 8 wk when CON skin was in telogen. RNA was purified from wound edges (1-2mm) of CON and KO skin (KO T) 1 day after wounding when the wound closure and re-epithelialization rate were faster in KO skin but collagen deposition in dermis was similar (Masson stain S2A Fig). The expression profiles of KO and CON wounds were obtained by microarray (GSE50670). When array data were applied to the TGFβ signaling pathway template in IPA software, the expression of TGFβ signaling related genes was generally reduced in KO wounds (S1 Fig lower panel green). IPA analysis also listed TGFβ as an upstream regulator (S1 Fig upper panel). TGFβ protein was detected in both epidermis (Fig 1H white triangle) and HF at the wounding edge (Fig 1H CON left panel). However, its level was substantially diminished in epidermis of KO wounds, although it remains comparable in the dermis (Fig 1H KO right panel). Enlarged images of H are focused on the HF/IFE area shown by dotted lines (Fig 1I CON, KO). Lack of green signals in KO is demonstrated by blue DAPI counter-staining, which is usually masked at green overlaid images (Fig 1I KO). The mRNA levels for epidermal TGFβ target genes (*Ctgf*) were substantially decreased in the KO in both normal and wounded skin at 1 day (S2B Fig). In contrast, the mRNA expression of several different myofibroblast markers including α-smooth muscle actin (α-SMA or acta2) were moderately reduced in normal and wounded skin of KO (S2C Fig), suggesting limited Med1/TGFβ regulation in myofibroblast driven wound contraction.

### *Med1* silencing reduces TGFβ signaling and increases cell proliferation *in vitro*

Next, we explored the mechanism of *Med1* dependent regulation of TGFβ signaling in cultured keratinocytes. They were maintained under pre-confluent conditions with low calcium to prevent their differentiation. *MED1* expression was blocked by transfection of siRNA (mixture of 4 *MED1* specific siRNAs) (siMED1), in which good blocking efficiency was confirmed by comparing with scrambled siRNA transfected control cells (siCon) (Fig 2A qPCR). *MED1* silencing decreased mRNA expression of ligands (*TGFβ1*, *TGFβ2*), signal transducer (*SMAD3*) and downstream target genes *CTGF*, *COL61* and *COL171* (Fig 2A). The activity of TGFβ signaling was evaluated by levels of nuclear phospho-SMAD2/3 (p-SMAD2/3) by using

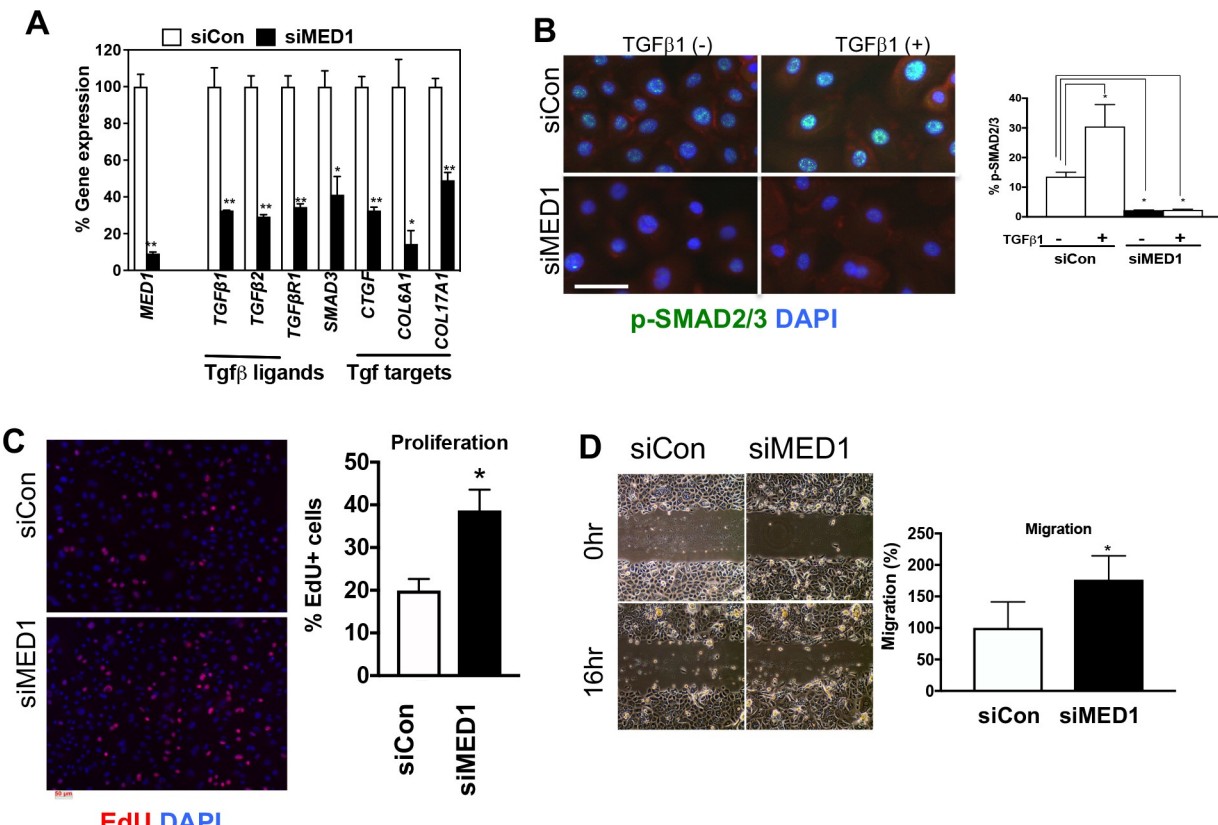

**Fig 2.** *MED1* **silencing suppresses TGFβ signaling** *in vitro*. (A) mRNA expression of *MED1*, *TGFβ* related genes in Med1 silenced (siMED1) and control (siCon) cells (qPCR, mean +/- SD n = 3, t-test, *p<0.05, **p<0.01). (B) Immuno-staining of p-SMAD2/3 (green) treated with or without rTGFβ1, red cytoplasmic β-catenin (red) with DAPI blue counterstain Bars = 50 um. Graph shows the number of nuclear p-SMAD2/3 positive cells (mean +/- SD, 3 fields, t-test, *p<0.05). (C) Cell proliferation evaluated by EdU (red). Graph shows quantitation of EdU positive cells per total cells (DAPI) (mean +/- SD, 5 fields, t-test, *p<0.05). (D) Photographs of monolayer culture of *MED1* silenced cells (siMED1) and control cells (siCon) right after scratch (0h) and overnight culture (16hr) are shown. The migration was quantitated by measuring empty spaces remaining after cell migration (5 fields, t-test, *p<0.05).

immunohistochemistry. *MED1* silencing reduced basal p-SMAD2/3 substantially (Fig 2B left) and suppressed recombinant TGFβ1 (rTGFβ1) induced expression (Fig 2B right) in keratinocytes. These results were supported by the quantitation of p-SMAD2/3 positive cells (Fig 2B right graph). The high level of p-SMAD2/3 without rTGFβ1 indicates that basal TGFβ activity is present in keratinocytes. *MED1* silencing also increased cell proliferation (Fig 2C EdU images with quantitation) and enhanced migration of keratinocytes in the *in vitro* scratch wound model (Fig 2D, images and quantitation). *MED1* silencing also changed cell morphology in which the honey cone epithelial shape is diminished (Fig 2D). These data demonstrate that *Med1* silencing reduces TGFβ signaling by suppressing expression of ligands (TGFβ1/β2) and signal transduction through p-SMAD2/3. Culture of mouse keratinocytes from adult Med1 KO and CON mice was not successful given the lack of adhesive and proliferative capability.

## MED1 is recruited into large enhancers/SEs in TGFβ1, TGFβ2 and SMAD3 loci

We then explored the underlying molecular mechanisms responsible for MED1 regulation of TGFβ signaling in cultured keratinocytes. We conducted chromatin immuno-precipitation

and next generation DNA sequencing (ChIP-seq) to define Med1 driven large enhancers, SEs, that are expected to drive transcription of cell specific genes [9]. Cells were fixed, sonicated and immune-precipitated using an antibody against MED1. Antibody against acetylated histone 3 at lysine 27 (H3K27ac) also mapped transcriptionally active enhancers co-related to SEs. CTCF antibody marks DNA looping positions to span SEs. MED1 bound DNAs were sequenced and analyzed as described [22, 26]. MED1 defined 9873 typical enhancers (Fig 3A) in keratinocytes that span short DNA segments (<1kb) proximal from transcriptional start sites (TSS). One example is shown near *JUNB* loci (Fig 3C). In contrast, MED1 retrieved 377 SEs (3.8% of total enhancers) (Fig 3A and 3B) that span long DNA segments (>20kb) with elevated MED1 peak heights, which were selected by Rose code as described previously [8, 12]. We also identified SE associated genes with annotations in an enhancer distribution profile (Fig 3B right side of dotted line). The *SMAD3* is listed having an SE in its loci in Fig 3D red bar with elevated MED1 peaks spanning a long DNA stretch (202 kb). The same SE is identified by high and broad H3K27ac peaks in the SMAD3 loci (Fig 3D). The *TGFβ2* loci also contained a SE (MED1 high peaks) spanning a long (89kb) DNA stretch (Fig 3E red bar). MED1 was also involved in a large enhancer at the *TGFβ1* loci, although it was not categorized as an SE by Rose code (S3 Fig black bar). We did not find SEs or large enhancers at loci for *TβRI*, *SMAD2*

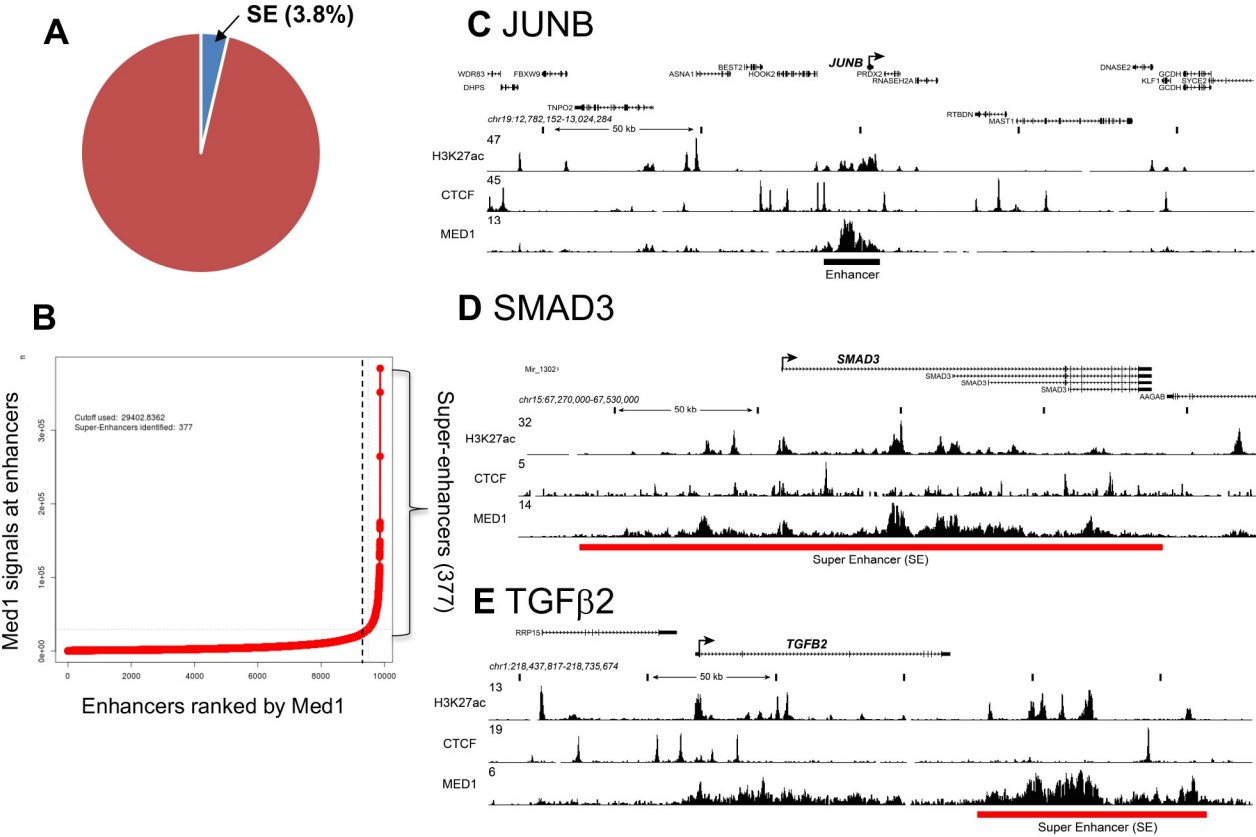

**Fig 3. ChIP-seq identifies SEs in regulatory regions of TGFβ signaling genes.** ChIP-seq was conducted on cultured keratinocytes using antibodies against MED1, H3K27ac and CTCF. (A) The number of SEs (3.8%) and typical enhancers (9873) identified by MED1 peak calling are depicted schematically. (B) Distribution of MED1 ChIP-seq signals (total reads) across the typical enhancers, in which enhancers are ranked by increasing MED1 signal. With a subset of enhancers, 377 SEs were identified that contain exceptionally high amounts of MED1 (right side of dotted line). (C-E) ChIP-seq binding profiles (reads per million per base pair) for the H3K27ac (active histone), CTCF (DNA loop site) and MED1 at *JUNB* (C), *SMAD3* (D), and *TGFβ2* (E) loci in keratinocytes. Red and black bar represent SE and typical enhancer, respectively. Schematic models of SE and typical enhancer are shown in Fig 6.

or TGFβ target genes (*CTGF*, *COL6A1*). These results suggest that MED1 directly facilitates transcription of the ligands (*TGFβ1/β2*) and their transducer (*SMAD3*), which are essential for TGFβ signaling. These data provide direct evidence for MED1 regulation of components of the TGFβ signaling pathway at the genome level. The role of SE associated genes other than genes in the TGFβ signaling pathway are beyond the scope of this study but will be elucidated in future studies.

## Pharmacological inhibition of TGFβ signaling enhances IFE and SG lineages

To further test whether TGFβ signaling is involved in maintenance of epidermal lineages, TGFβ signaling was locally inhibited in skin by RepSox (RS), a selective inhibitor of TGFβ receptors (TGFβR1/TGFβR2). TGFβ signaling was inhibited locally in the skin by topical application of a small volume of drug (RS), limiting systemic effects from oral dosing. The RS (30 μl) was applied to the surface of back skin (shaved) during telogen of C57BL6 mice (8–9 wk). The drug was applied to the skin daily for 5 days (5 d) or for 12 days (1, 2, 3, 4, 5, 8, 9, 10, 11, 12th day) (12d). RS application increased the thickness of IFE at 5d leading to hyperplasia at 12d (Fig 4A HE yellow arrows, S4C Fig quantitated thickness). RS increased cell proliferation in the isthmus/JZ/I (Fig 4C red triangles) and basal cells of the IFE by 5d treatment as shown by images (Fig 4D red triangles), that is supported by quantitation of PCNA positive cells (S4C Fig graph PCNA). Moreover, RS increased the IFE markers KRT1, IVL, FLG in the upper HF at 5d (Fig 4B RS 5d red triangles) and throughout the abnormally enlarged HFs at 12d (Fig 4B RS 12d). RS treatment also enhanced SG differentiation as demonstrated by 1) increased size of SG (Fig 4A 5d red triangle), 2) increased Oil Red O reactive lipids in SG (Fig 4E RS red triangles) (S4C Fig quantitation), and 3) increased PPARγ positive cells in SG (Fig 4F white triangles) by 5d treatment, the quantitation of which is shown in S4C Fig (PPARγ). However, SG differentiation was not increased at 12d. The mRNA levels of SG/IFE markers were increased at 5 d (Fig 4G) consistent with the histological results (Fig 4B, 4E and 4F). In contrast, RS blunted hair differentiation as mRNA expression of the hair markers of *Krt31*, *Krt32*, *Krt2-16* was decreased (Fig 4G). We confirmed efficient inhibition of TGFβ signaling by RS by showing reduced *Ctgf* expression (Fig 4G).

## Pretreatment of the skin with RepSox accelerates wound re-epithelialization

The skin was pretreated with RS for 5 d following which a 3 mm skin biopsy was taken as previously described [29]. Wounds in RS pretreated skin closed faster compared to wounds made in non-treated skin at both 1d and 3d after wounding as shown by morphology (Fig 5A, S4A Fig) and size of wounds (Fig 5B quantitation). The effects were reproduced with statistical significance. The RS pretreatment accelerated the re-epithelialization rate as shown by representative HE images (Fig 5C, S4B Fig) and quantitation (Fig 5D). The RS treatment increased cell proliferation over 10 fold in isthmus/JZ/I (Fig 5E RS 0d red arrows) in non-wounded skin, and the increase was sustained in isthmus/JZ/I after wounding at 1 d (Fig 5E RS 1d). We did not observe these effects when RS was applied post-wounding, discounting a role of RS on post wounding inflammation and fibrosis with respect to re-epithelialization. Our RS results (5d) are consistent with the phenotypes of *Med1* KO skin in that both show increased proliferation, enhanced IFE/SG lineages [13], and accelerated wound re-epithelialization [16]. However, accelerated healing in Med1 KO or RS treated skin may not be solely due to re-epithelialization. Nor can we exclude other as yet unidentified or untested factors also altered by Med1 KO

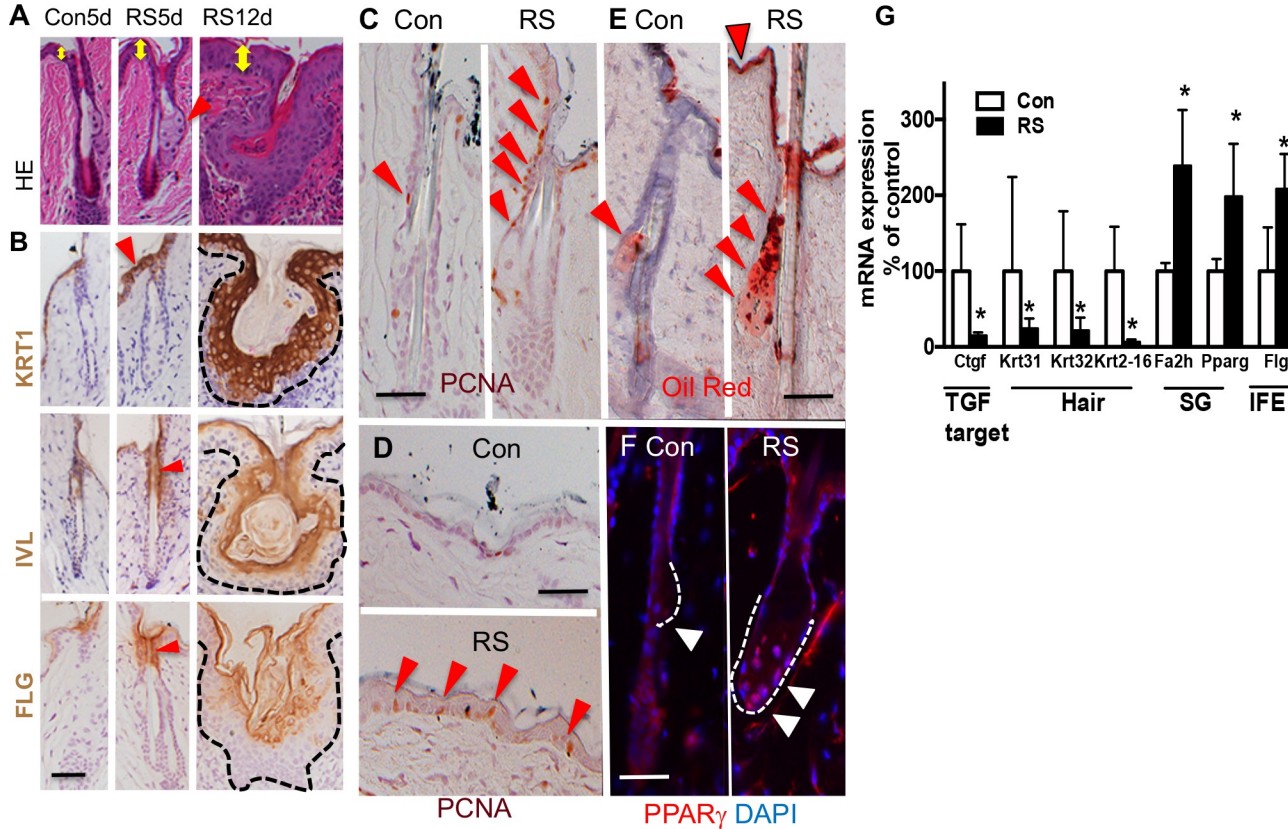

**Fig 4. TGFβ inhibitor enhances epidermal lineages for IFE and SG.** (A) H&E staining of skin that are treated with TGFβR inhibitor, RepSox (RS), for 5d and 12d. Yellow arrows show epidermal thickness. (B) Immunostaining of RS treated skin (5d and 12d) for IFE markers (KRT1, IVL, FLG) (brown with blue counter staining). Red arrows show the position of IFE markers at 5d. (C, D) PCNA staining (brown with blue counterstaining). PCNA positive cells are shown by red triangles in HF (C) and IFE (D) at 5d after RS treatment. (E) Oil Red O staining (red) with blue counterstain for 5d Con and RS treated skin. The triangles show increased lipids accumulated in IFE (top of the skin) and SG (lower area), in RS skin. The shape of the SG is shown by dotted lines. (F) The PPARγ staining showing the shape of the SG with dotted line. The RS skin has enlarged SG (red with blue DAPI counterstaining). Red signal in dermis is non-specific. (G) mRNA expression of lineage specific markers for hair, SG and IFE in RS treated skin compared to control skin (5d) (qPCR mean +/- SD, n = 3, t-test, *p<0.05).

(as listed in S1 Table), that might contribute to the rapid wound closure in either *Med1* KO or RS pretreated skin.

The similarity between the phenotypes of *Med 1* KO skin and RS treated skin with respect to their impact on epidermal lineages suggests that TGFβ signaling mediates at least part of Med1 effects on epidermis. For example, *Med1* may facilitate an anti-proliferative function of TGFβ1 in IFE/SG potentially through the production of ECM related components such as CTGF or COL6A1. *CTGF*, the most down-regulated gene in *Med1* KO skin, may be involved in such regulation as previously proposed [30]. These ECM related genes may maintain a microenvironment of epidermal cells fostering cell quiescence and suppressing IFE/SG differentiation as seen in other tissues [17, 19]. In contrast, TGFβ2 secreted from the hair germ may promote HF development by counteracting quiescent BMP signaling as previously described [31].

The suggestion that TGFβ signaling contributes to these actions of Med1 in skin is further supported by several TGFβ related transgenic mouse models. Increased SG/IFE differentiation with hair defects was reported in inducible dominant negative *TGFbR2* null skin [32]. Hair defects are observed in conditional Krt14 driven *TGFbR2* null skin [31], and in mice lacking

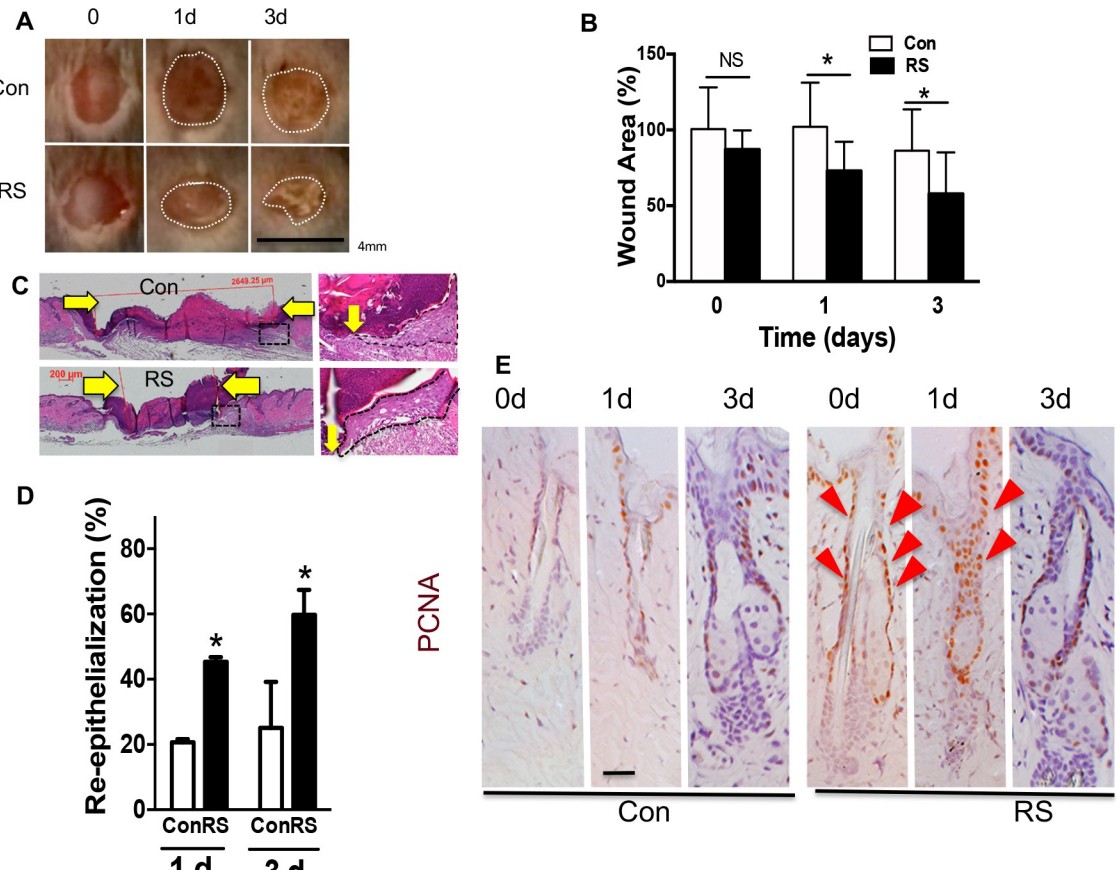

**Fig 5. TGFβ inhibitor RS accelerates wound re-epithelialization.** (A) Photographs of skin wounds at 0d, 1d, and 3d after 3mm biopsy was excised from RS pretreated or control skin. *Bar* = 4mm. Variation in 2 wounds and 3 mice are in Fig S4A Fig. (B) Quantitated wound size (mean +/- SD *p<0.05, 2 sections per mouse, 3 mice each group). (C) H&E staining of skin wounds (3d) with enlarged images (right). Yellow arrows show wound edges. (D) Re-epithelialization rate at 1d and 3d (mean +/- SD, n = 6, *p<0.05) that is calculated by the ratio of the distance between the wounding edges of the epithelial tongues to the original diameter of the wound. (E) Cell proliferation at wound edges. PCNA staining (brown) with blue counterstaining. Red triangles show brown PCNA positive cells increased in RS treated wounds. *Bars* = 50 um. Representative images of reproduced results are shown.

desmosomal component *Col17a1*, by which hair and melanocyte stem cells are regulated by *TGFβ* signaling [33]. Accelerated skin wound healing phenotypes were observed in *SMAD3*-null mice [34]. Conditional *TGFβR2* null mice [35] are noted to have enhanced wound re-epi-thelialization by increasing cell proliferation and migration, that are also observed in our siMed1 keratinocytes (Fig 2). These reports support our conclusion that Med1 regulation of skin lineages and wound re-epithelialization involves modulation of TGFβ signaling. That said we recognize that other pathways such as the wnt/β-catenin or other pathways may also be involved, but that remains to be elucidated.

Our results suggest that the depletion of Med1 from Krt14 expressing keratinocytes acceler-ates wound healing via increased re-epithelialization. However, we do not exclude unidentified mechanisms that contribute to the rapid wound closure in either *Med1* KO or RS pre-treated skin. One possibility is that the Med1/TGFβ signaling indirectly affect dermal myofibroblasts, that drives wound contraction to accelerate wound closure [36]. However, our results showed the changes are minor or even in opposition as myofibroblast markers were reduced in both normal and wounded skin of KO. Therefore, it is unlikely that accelerated would closure is due to the contraction. In addition, the same may be true in RS pre-treated skin because other

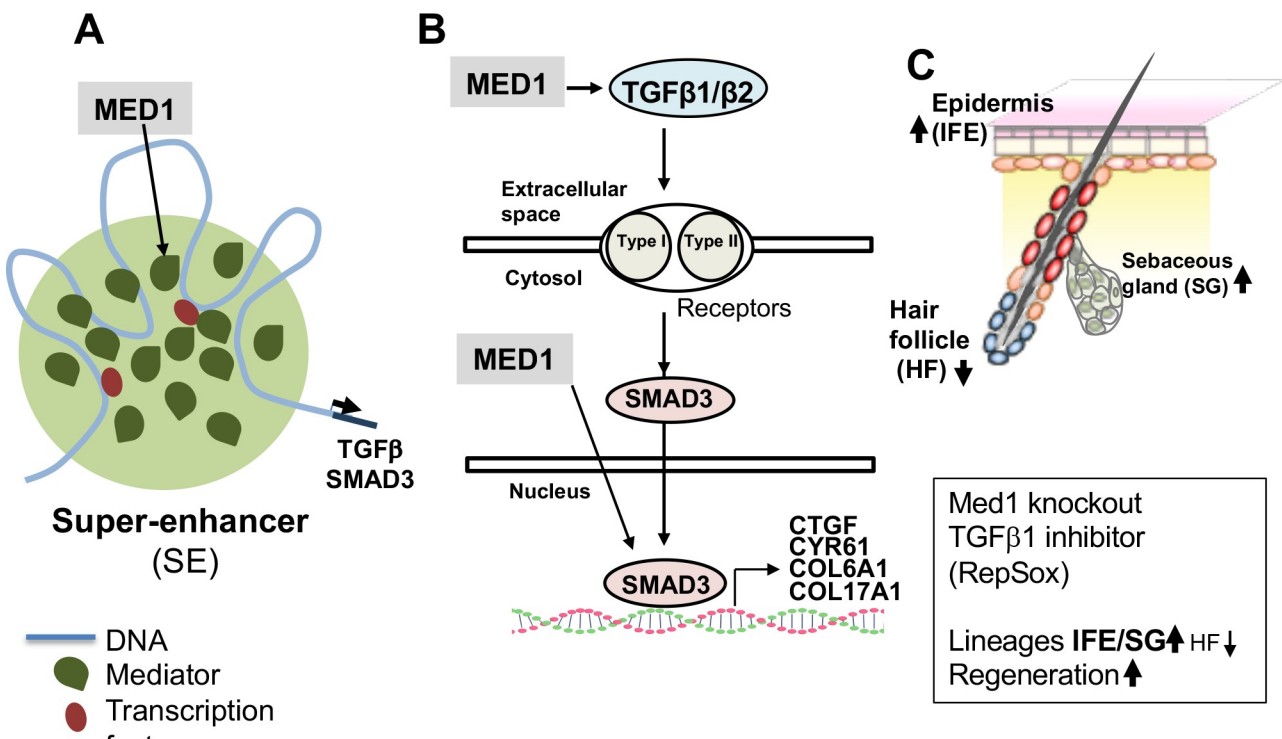

**Fig 6. A proposed model in which *Med1* regulates TGFβ signaling, epidermal lineages and regeneration.** (A) *Med1* mediates transcription of *TGFβ1/β2* and *SMAD3*. Mediator (green) including *Med1* subunit is densely incorporated into large super-enhancers (SE) that drive selected genes involved with lineage maintenance and tissue regeneration [9]. (B) *Med1* induces TGFβ1/β2 and SMAD3 (arrows) to support TGFβ downstream target gene expression (e.g. *CTGF*, *COL6A1*) that may maintain cell quiescence or epidermal lineages. (C) *Med1* null skin increases epidermal lineages but suppresses HF lineage. Med1 and TGFβ signaling may have distinct but overlapping roles in regulating different lineages as described in results and discussion. TGFβ inhibition by RepSox (5d) recapitulates the phenotypes of *Med1* null skin with respect to accelerated wound re-epithelialization, providing a potential new approach to treating diseases such as delayed wound healing.

study showed that topical application of an TGFβ antagonist peptide reduces wound contraction [37]. More detailed mechanism for regulation of wound healing by Med1/TGFβ will be elucidated by our future investigation.

## Conclusions

Our study demonstrates that *Med1* regulates TGFβ signaling and controls epidermal lineages and regeneration as illustrated by our proposed model (Fig 6). *MED1* (green*)* may mediate transcription of *TGFβ1/β2* and *SMAD3* via the formation of large super-enhancers (SE) (Fig 6A). *Med1* targets TGFβ1/β2 and SMAD3 (arrows) to induce TGFβ downstream target genes (e.g. *CTGF*, *COL6A1*) that may maintain cell quiescence and/or epidermal lineages (Fig 6B). *Med1* null skin or TGFβ inhibition by RepSox (5d) enhance epidermal lineages towards IEF and SG and accelerate epidermal regeneration during cutaneous wound healing (Fig 6C). Our findings present a potential new approach to accelerate injury induced epidermal regeneration by blocking Med1 or TGFβ signaling to treat poorly healing wounds.

## Supporting information

**S1 Table. The TGFβ1 is listed as an upstream regulator for KO keratinocytes.**
(XLSX)

**S1 Fig. TGFβ signaling pathway is inhibited in KO wounded skin (1d).**
(TIF)

**S2 Fig. Med1 deletion affects wounded skin 1 day after injury.**
(TIF)

**S3 Fig. MED1 forms large enhancer at the TGFβ1 loci.**
(TIF)

**S4 Fig. TGFβ1 inhibitor RepSox accelerates wound re-epithelialization rate.**
(TIF)

**S1 File.**
(DOCX)

## Acknowledgments

We thank Ms A. Menendez, Ms W. Mayer for their assistance in maintaining the mice. We are also grateful to Dr L. Hu, Mr C. Fong, Ms M. Zhang, and Dr SHK Wong for technical support.

## Author Contributions

**Conceptualization:** Yuko Oda, Akiko Hata, J. Wesley Pike, Daniel D. Bikle.

**Data curation:** Yuko Oda, Mark B. Meyer, J. Wesley Pike.

**Formal analysis:** Yuko Oda, Thai Nguyen, Mark B. Meyer.

**Funding acquisition:** Yuko Oda, Daniel D. Bikle.

**Investigation:** Yuko Oda, Thai Nguyen, Mark B. Meyer, J. Wesley Pike, Daniel D. Bikle.

**Methodology:** Yuko Oda, Thai Nguyen, Akiko Hata, Mark B. Meyer, J. Wesley Pike.

**Resources:** Yuko Oda, Mark B. Meyer.

**Software:** Thai Nguyen, Mark B. Meyer.

**Supervision:** Yuko Oda, J. Wesley Pike, Daniel D. Bikle.

**Validation:** Yuko Oda, Thai Nguyen, Akiko Hata, Mark B. Meyer, J. Wesley Pike.

**Visualization:** Yuko Oda, Thai Nguyen, Mark B. Meyer, J. Wesley Pike.

**Writing – original draft:** Yuko Oda.

**Writing – review & editing:** Yuko Oda, Akiko Hata, Daniel D. Bikle.

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
