## [Decision Letter · Decision Letter 0]

23 Jun 2020

PONE-D-20-11557

Deletion of Mediator 1 suppresses TGFβ signaling leading to changes in epidermal lineages and regeneration

PLOS ONE

Dear Dr. Oda,

Thank you for submitting your manuscript to PLOS ONE. After careful consideration, we feel that it has merit but does not fully meet PLOS ONE’s publication criteria as it currently stands. Therefore, we invite you to submit a revised version of the manuscript that addresses the points raised during the review process.

We look forward to receiving your revised manuscript.

Kind regards,

Roberto Mantovani

Academic Editor

PLOS ONE

Journal Requirements:

2. We note that you are reporting an analysis of a microarray, next-generation sequencing, or deep sequencing data set. PLOS requires that authors comply with field-specific standards for preparation, recording, and deposition of data in repositories appropriate to their field. Please upload these data to a stable, public repository (such as ArrayExpress, Gene Expression Omnibus (GEO), DNA Data Bank of Japan (DDBJ), NCBI GenBank, NCBI Sequence Read Archive, or EMBL Nucleotide Sequence Database (ENA)). In your revised cover letter, please provide the relevant accession numbers that may be used to access these data. For a full list of recommended repositories, see http://journals.plos.org/plosone/s/data-availability#loc-omics or http://journals.plos.org/plosone/s/data-availability#loc-sequencing.

3. To comply with PLOS ONE submissions requirements, in your Methods section, please provide additional information on the animal research and ensure you have included details on (1) methods of sacrifice, (2) methods of anesthesia and/or analgesia, and (3) efforts to alleviate suffering.

4. In your Methods section, please include a comment about the state of the animals following this research. Were they euthanized or housed for use in further research? If any animals were sacrificed by the authors, please include the method of euthanasia and describe any efforts that were undertaken to reduce animal suffering.

5. In your Methods section, please provide additional details regarding each of the cell lines used in your study, including any quality control testing procedures, and ensure you have described the source. For more information regarding PLOS' policy on materials sharing and reporting, see https://journals.plos.org/plosone/s/materials-and-software-sharing#loc-sharing-materials, and for more information on PLOS ONE's guidelines for research using cell lines, see https://journals.plos.org/plosone/s/submission-guidelines#loc-cell-lines.

Reviewers' comments:

Reviewer's Responses to Questions

**Comments to the Author**

1. Is the manuscript technically sound, and do the data support the conclusions?

Reviewer #1: Yes

2. Has the statistical analysis been performed appropriately and rigorously? 

Reviewer #1: Yes

3. Have the authors made all data underlying the findings in their manuscript fully available?

Reviewer #1: Yes

4. Is the manuscript presented in an intelligible fashion and written in standard English?

Reviewer #1: Yes

5. Review Comments to the Author

Reviewer #1: Oda and coworkers have explored the molecular mechanisms underlying the phenotype of Med1 KO mice that show increased wound healing and an alteration in ectodermal lineage differentiation. By means of gene expression arrays, chromatin immunoprecipitation and in vitro assays they showed that Med1 is a transcriptional regulator associated to super-enhancers that regulate the expression of TGFbeta ligands and the TGFbeta singal tranducer SMAD3. They linked the phenotype of Med1KO mice to the activity of TGFbeta SMAD3 signaling pathway. Inhibition of Med1 by gene deletion or siRNA transfection alters the differentiation of epidermal cells favouring interfollucular epidermis and sebaceous gland differentiation and contrasting hair follicle differentiation. The same phenotype is obtained in vivo by topically applying TGFbeta Receptor inhibitor RepSox.

The Manuscript is well written and the experiments carefully controlled. My main concern is related to the fact that the whole mechanism described here seems to be cell autonomous while, particularly for TGFbeta signaling in wound healing there is an high level of interaction between epidermal cells and dermal fibroblasts that have been overlooked in this paper.

6. PLOS authors have the option to publish the peer review history of their article (what does this mean?). If published, this will include your full peer review and any attached files.

Reviewer #1: No

---

## [Author Response · Author response to Decision Letter 0]

24 Jul 2020

Response to reviewers

“Deletion of Mediator 1 suppresses TGFβ signaling leading to changes in epidermal lineages and regeneration” (PONE-D-20-11557)

We appreciate your positive evaluation of our manuscript. We have carefully reviewed the reviewer’s comments and journal requirements. We revised our manuscript by addressing one comment related to dermal fibroblasts by providing new data and further discussion. Our changes in the text is shown by underlines.

Reviewer’s comments and journal requirements are shown by italics. We show our answers afterwards.

Reviewer #1: Oda and coworkers have explored the molecular mechanisms underlying the phenotype of Med1 KO mice that show increased wound healing and an alteration in ectodermal lineage differentiation. By means of gene expression arrays, chromatin immunoprecipitation and in vitro assays they showed that Med1 is a transcriptional regulator associated to super-enhancers that regulate the expression of TGFbeta ligands and the TGFbeta singal tranducer SMAD3. They linked the phenotype of Med1KO mice to the activity of TGFbeta SMAD3 signaling pathway. Inhibition of Med1 by gene deletion or siRNA transfection alters the differentiation of epidermal cells favouring interfollucular epidermis and sebaceous gland differentiation and contrasting hair follicle differentiation. The same phenotype is obtained in vivo by topically applying TGFbeta Receptor inhibitor RepSox.

The Manuscript is well written and the experiments carefully controlled. My main concern is related to the fact that the whole mechanism described here seems to be cell autonomous while, particularly for TGFbeta signaling in wound healing there is an high level of interaction between epidermal cells and dermal fibroblasts that have been overlooked in this paper.

Answer: We appreciate reviewer’s positive evaluation. About the last comment, we totally agree that TGFβ signaling regulates both epidermal keratinocytes and dermal fibroblasts and their interaction during cutaneous wound healing, although these regulations may be context dependent.

We would like to point out that our study focuses on the mechanism of Med1 action in epidermal keratinocytes by deleting the Med1 gene specifically in Krt14 expressing keratinocytes but not from dermal fibroblasts. In addition, we here report the Med1 dependent regulation of TGFβ regulation in epidermal lineages and wound re-epithelialization but not complexed cutaneous wound healing itself.

However, we revised our manuscript as we respect reviewer’s last comment.

1. In the revised manuscript, we added new data demonstrating the changes in dermis. We have already mentioned this topic briefly without data (data not shown) but now we added data. 

a. Med1 KO wounds have minor changes in collagen deposition compared to control wounds (Masson staining in Supplemental S2 A Fig).

b. Med1 deletion in epidermis moderately decreased the mRNA expression of myofibroblast markers in both non wounded and wounded skins (Supplemental S2 C Fig).

2. We add discussion about potential mechanisms to accelerate wound closure in Med1KO and RS-pretreated skin by referring published papers as follows. 

Our results suggest that the depletion of Med1 from Krt14 expressing keratinocytes accelerates wound healing via increased re-epithelialization. However, we do not exclude unidentified mechanisms that contribute to the rapid wound closure in either Med1 KO or RS pre-treated skin. One possibility is that the Med1/TGF� signaling indirectly affect dermal myofibroblasts, that drives wound contraction to accelerate wound closure [36]. However, our results showed the changes are minor or even in opposition as myofibroblast markers were reduced in both normal and wounded skin of KO. Therefore, it is unlikely that accelerated would closure is due to the contraction. In addition, the same may be true in RS pre-treated skin because other study showed that topical application of an TGF� antagonist peptide reduces wound contraction [37]. More detailed mechanism for regulation of wound healing by Med1/TGF��will be elucidated by our future investigation.

36. Lichtman MK, Otero-Vinas M, Falanga V. Transforming growth factor beta (TGF-beta) isoforms in wound healing and fibrosis. Wound Repair Regen. 2016;24(2):215-22. Epub 2015/12/26. doi: 10.1111/wrr.12398. PubMed PMID: 26704519.

37. Huang JS, Wang YH, Ling TY, Chuang SS, Johnson FE, Huang SS. Synthetic TGF-beta antagonist accelerates wound healing and reduces scarring. FASEB J. 2002;16(10):1269-70. Epub 2002/08/03. doi: 10.1096/fj.02-0103fje. PubMed PMID: 12153996.

These data and references may not completely address the reviewer’s last comment. We believe these subjects remains for our future studies.

Journal Requirements:

 Answer: We carefully revised our manuscript to matched to PLOS ONE style. The revised part is shown by underlines. 

2. We note that you are reporting an analysis of a microarray, next-generation sequencing, or deep sequencing data set. PLOS requires that authors comply with field-specific standards for preparation, recording, and deposition of data in repositories appropriate to their field. Please upload these data to a stable, public repository (such as ArrayExpress, Gene Expression Omnibus (GEO), DNA Data Bank of Japan (DDBJ), NCBI GenBank, NCBI Sequence Read Archive, or EMBL Nucleotide Sequence Database (ENA)). In your revised cover letter, please provide the relevant accession numbers that may be used to access these data. For a full list of recommended repositories, see http://journals.plos.org/plosone/s/data-availability#loc-omics or http://journals.plos.org/plosone/s/data-availability#loc-sequencing.

Answer: We have already showed accession numbers for microarray data, and now added the number for Chip-seq data (GSE154221) as shown below.

Accession Numbers 

The microarray and Chip-seq data were submitted to a public database (GEO/NCBI/NIH http://www.ncbi.nlm.nih.gov/geo). Microarray data for Med1 KO epidermal cells (10 wk), KO whole skin (10 wk), and KO skin wounds (1d after wounding) are available with accession numbers GSE50672, GSE50671, GSE50670, respectively under the super-series GSE50673. The ChIP-seq data for keratinocytes immunoprecipitated with H3K27ac, CTCF and Med1 antibodies are available with accession number GSE154221. 

3. To comply with PLOS ONE submissions requirements, in your Methods section, please provide additional information on the animal research and ensure you have included details on (1) methods of sacrifice, (2) methods of anesthesia and/or analgesia, and (3) efforts to alleviate suffering.

4. In your Methods section, please include a comment about the state of the animals following this research. Were they euthanized or housed for use in further research? If any animals were sacrificed by the authors, please include the method of euthanasia and describe any efforts that were undertaken to reduce animal suffering.

Answer: We added underlined sentences in the method section (Supplemental information) to answer comments 3 and 4. 

Skin wounding protocol

Male mice for Med1 KO and CON (n=3-6 each) were used for wounding studies. They were studied at 8-10 wk of age when CON skin was mostly pink (telogen) as wound healing is related to HF cycling (Ansell et al., 2011). Mice are anesthetized with isoflurane inhalation using vaporizer. After shaving hair, two 3 mm full thickness skin biopsies were taken from the upper portion of their backs. Respiratory rate is monitored and adequacy of anesthesia will be confirmed by lack of response to foot pinch or whisker stimulation during the procedure. After surgery, mice are kept in warm pads for their recovery from anesthesia. All efforts were made to minimize suffering. As KO skin always contained abnormal black/grey anagen patches mixed in pink telogen skin, skin biopsies were taken from both anagen and telogen skin only in the case of KO mice. The hair cycle was judged by visual inspection of skin color. The wounds were monitored during their recovery as described (Oda et al., 2015). Skin wounds (3mm) were harvested after 1 or 3 d of wounding following euthanasia done by decapitation under overdose of isoflurane. All the mice are sacrificed after wounding studies and are not use for other purposes. For analyses, one of the wounds was used for histology and the other for mRNA levels. The rim of tissue around the other wound in the same mouse was harvested and stored in RNA later for subsequent mRNA expression analyses. 

5. In your Methods section, please provide additional details regarding each of the cell lines used in your study, including any quality control testing procedures, and ensure you have described the source. For more information regarding PLOS' policy on materials sharing and reporting, see https://journals.plos.org/plosone/s/materials-and-software-sharing#loc-sharing-materials, and for more information on PLOS ONE's guidelines for research using cell lines, see https://journals.plos.org/plosone/s/submission-guidelines#loc-cell-lines.

Answer: We used primary cells but not any cell lines in this study. 

 Answer: We provided three data in supporting information and removed the phrase “data not shown”.

a. IPA microarray data of Med1 wounds to show TGF� is an upstream regulator (Supplemental S1 Fig upper panel)

b. Masson staining data for control wounds and Med1 wounds (Supplemental S2 A Fig).

c. mRNA expression of �-SMA (acta2) and other myofibroblast makers in non-wounded and wounded skin of Med1 KO and CON (Supplemental S2 B Fig). 

We removed the phrases that refer dermal changes in RS pre-treated skin (line 351). 

7. Please include captions for your Supporting Information files at the end of your manuscript, and update any in-text citations to match accordingly. Please see our Supporting Information guidelines for more information: http://journals.plos.org/plosone/s/supporting-information

Answer: We added the cations (titles) for Supplemental figures at the end of the manuscript.

We revised the manuscript by making minor changes as follows.

a. The numbering of supplemental figures is changed. As former S1-1 Fig is Table, it became S1 Table. Then, we removed the sub-numbering system such as 1-1, 1-2, 3-1 etc.

b. The figure legends for supplemental figures are shortened to fit to one line to meet your journal requirements.

---

## [Editor Report · Decision Letter 1]

10 Aug 2020

Deletion of Mediator 1 suppresses TGFβ signaling leading to changes in epidermal lineages and regeneration

PONE-D-20-11557R1

Dear Dr. Oda,

We’re pleased to inform you that your manuscript has been judged scientifically suitable for publication and will be formally accepted for publication once it meets all outstanding technical requirements.

Kind regards,

Roberto Mantovani

Academic Editor

PLOS ONE
---

## [Editor Report · Acceptance letter]

13 Aug 2020

PONE-D-20-11557R1 

Deletion of Mediator 1 suppresses TGFβ signaling leading to changes in epidermal lineages and regeneration 

Dear Dr. Oda:

I'm pleased to inform you that your manuscript has been deemed suitable for publication in PLOS ONE. Congratulations! Your manuscript is now with our production department. 

Kind regards, 

on behalf of

Prof. Roberto Mantovani 

Academic Editor

PLOS ONE